# Dimensionality of the Causes of Churning: A Multivariate Statistical Analysis

Olga Alexandra Chinita Pirrolas [1,*] and Pedro Miguel Alves Ribeiro Correia [2,*]

1    Institute of Social and Political Sciences, University of Lisbon, 1300-663 Lisbon, Portugal
2    Faculty of Law, University of Coimbra, 3004-528 Coimbra, Portugal
*    Correspondence: olgaalexandrap@gmail.com (O.A.C.P.); pcorreia@fd.uc.pt (P.M.A.R.C.)

**Abstract:** The present study was conducted in Portugal and had, as its object of study, workers from Portuguese companies belonging to several sectors of activity. The main goal of this study was the identification of the dimensions related to the causes of churning and to analyse its applicability in the management of human resources to promote individual and corporate welfare. Its specific targets were (a) to make a sociographic characterisation of the workers; (b) to make their professional characterisation; (c) to analyse the perception workers had in relation to the selected dimensions under study. Through the gathering of data per questionnaire, a sample consisting of 801 answers was considered. First, we resorted to a multivariate statistical analysis through the application of an Exploratory Factor Analysis (EFA) that allowed for the selection of the most relevant dimensions, followed by a descriptive statistical analysis on the collected sample and used items. Finally, we resorted to a TwoStep Cluster analysis that allowed for the identification of two Clusters of workers with a differentiated probability for the occurrence of churning.

**Keywords:** churning; individual and corporate welfare; dimensionality of churning; causes of churning; TwoStep cluster analysis

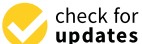



## 1. Introduction

The promotion of organisational and individual well-being is a subject that has been gaining relevance aiming at providing workers better conditions on both personal and professional levels [1].

Organisations are characterised for the way their processes are structured and managed. Their characteristics are crucial for outlining and implementing policies and procedures that positively affect the health and well-being of its workers who, in turn, contribute with a sense of commitment for the overall effectiveness of the organisation.

Resulting from the latest ongoing organisational changes derived from mergers, privatisations, and outsourcing, workers have been asked all sorts of adaptative skills, as is the case of the implementation and adaptation to new technologies and the flexibility in activities and competencies.

Thus, in the context of economic globalisation and of social and political complexities derived from scientific and technological innovations, organisations have felt the need to adapt to it by changing its policies and procedures to skirt the pressing issue of how to ensure the permanence of its workers in the organisation [2]. This problem results from competition, that is, from the ongoing competitiveness between organisations aiming to attract the most competent, experienced, and qualified workers [3]; hence, the churning of human resources constitutes one of the main problems when talents decide to leave organisations [4]. In this context, the phenomenon of churning of human resources arises. Despite the multiplicity of definitions of the latter and its relation to turnover (hirings and departure of workers), this article took as its premise the costs associated with the voluntary exits of workers, by focusing on the costs derived from such substitutions [5].

For the above reasons and despite being a little explored concept both in literary and operational terms, the concept of churning of human resources has been gaining visibility in the field of human resources due to increasing competitiveness in the labour market, leading workers to make the decision to voluntarily leave the organisation by accepting offers made from other competitors [6].

The relevance of this study is based on the opportunity to transpose to human resources management the reality experienced by workers in organisations and, thus, promote the importance of operationalising the concept of churning, still a seldomly applied and unfamiliar concept, in order to contribute to talent retention to minimise the costs associated with new hires, as well as to contribute to the dissemination and development of a very complex and little explored topic in Portugal.

Taking the above into consideration, this article has, as its general objective, the identification of the dimensions related to the causes of churning and the analysis of its applicability in the management of human resources to promote individual and corporate welfare.

Based on the foregoing considerations, a brief review of the literature follows by addressing the concept of churning of human resources; the dimensions of the main causes of churning; the role the management of human resources has in attenuating churning to promote individual and organisational well-being.

Regarding our chosen method, through the results retrieved from the performed questionnaire, we resorted to a multivariate statistical analysis aiming to select the most relevant dimensions, to make a sociographic characterisation of the survey respondents, and, finally, we resorted to the TwoStep Cluster analysis to identify groups of workers with differentiated probabilities for the occurrence of churning.

From the obtained results retrieved from the statistical analysis, we intended to conjecture which actions should be taken by organisations to promote organisational and individual well-being and to keep the best workers in the organisation.

### 1.1. Churning of Human Resources

Due to the economic context in which the labour market is integrated and considering the expectations of each worker, the decision to stay or leave an organisation one works for to meet its needs is a matter of right of freedom [5]. Although churning is related to turnover, or considered to be an aspect of turnover, there are some differences. While turnover's primary focus is related to the turnover of workers (entries and exits), and although churning is also related to it, this concept has, as its main focus, the costs associated with worker replacements derived from voluntary exits, that is, churning is only related to the costs of hires that stem from such replacements [7].

However, churning of human resources is categorised by two types: voluntary and involuntary churning. Voluntary churning is described as a worker's decision to leave an organisation, whilst involuntary churning is derived from an organisation's decision to lay-off a worker [8].

Based on the foregoing considerations, this article focuses on voluntary churning as it is considered the most problematic for organisations, oftentimes associated with negative reasons: conflicts with supervisors or colleagues, lack of recognition, lack of motivation, uninteresting work, lack of professional training, scarce career opportunities, low wages, poor working conditions, etc. [9].

In this sense, organisations tend to focus on workers' voluntary resignations as these lead to losses and associated costs stemming from the departure of skilled workers in whom organisations had invested and that still chose to leave due to better working conditions found elsewhere [10].

In brief: considering the above and the multiplicity of definitions and the lack of consensus regarding the definition of churning, notwithstanding its interconnectedness with turnover (entries and exits of workers), in this article, we chose to identify it as the costs resulting from voluntary departures, i.e., the relation established between churning of human resources and the replacements resulting from these departures [11]. Assuming

such a definition and its categories, this study focused on voluntary churning, as it is an issue with which organisations are constantly being confronted with nowadays, i.e., with the loss of their best employees and the ensuing costs resulting from the latter and new hires to cope with needed replacements [12,13].

### 1.2. Dimensionality of the Causes of Churning

Based on the revision of literature and state of the art studied from empirical research, the present study aimed to identify all dimensions related to the causes of churning and analyse its applicability in the management of human resources to provide for individual and organisational well-being.

Our method for the selection of literature to be analysed was possible by resorting to the following databases: Researchgate; Google Scholar; B-on; ScienceDirect (Elsevier, Amsterdam, The Netherlands); Wiley Online Library; Scielo; Sage Journals Online. The option for the selection of the dimensions of the causes of churning fell upon the frequency in which these surfaced within the consulted literature, having been considered as the most relevant: work environment; leadership; recognition; schedule; salary; career progression; responsibility. A brief description of each dimension follows, with emphasis on the relevancy each one has for organisations' overall satisfaction and well-being, contributing for the reduction in churning.

### 1.2.1. Work Environment

Work environment can be defined as a set of factors comprised by working conditions, adequate remuneration, equal employment opportunities, etc., generating a constructive environment where workers can apply their skills and abilities to meet their interests, providing for corporate and individual welfare in the workplace [14].

Horwitz et al. [15] stated that a good work environment is a factor conducive to employee retention. Organisations in competitive settings must use various strategies linked to human resource practices to improve their work environment [16].

### 1.2.2. Leadership

Akdol and Arikboga [17] considered that leaders should contribute to the personal development and empowerment of their employees. Their aim should be to encourage workers to be more autonomous, having as their main task the ability to motivate employees and teams, contributing to their well-being and permanence within the organisation.

Several studies have showed that leadership styles have a great impact on employee retention within organisations, resulting in a decrease in the occurrence of churning. A leader's conduct has a direct impact on the commitment an employee will have towards the organisation, highlighting the influence it has on their retention [18,19]. Ref. [20] reported that leadership styles impact the occurrence of churning due to an unbalanced treatment of workers, requiring leaders to question themselves about their personal judgments, biases, and assumptions, ultimately resulting in increased motivation and organisational effectiveness.

### 1.2.3. Recognition

Recognition can be defined as the appreciation or approval of achievements or positive behaviours given to an employee or a team, which can be demonstrated by giving a compliment, a personal reference for their performance, or even through small gestures that are important to employees, enhancing their well-being in the organisation [21].

Recognition results from a set of human resources' practices used by organisations, namely, performance evaluations. In this sense, the employer and employee benefit as it helps employees understand their strengths and weaknesses and, based on the provided feedback regarding their performance, this can lead to increased monetary benefits (remuneration) as well as non-monetary benefits such as recognition. It may even lead to career development; thus, it is an important factor in determining the occurrence of

churning, i.e., it is necessary to emphasise recognition to motivate employees to stay in the organisation [22].

### 1.2.4. Schedule

The possibility to choose companies that allow for flexible working hours allows workers to better adapt to their work, to better manage their workload and to manage responsibilities around their personal lives [23].

Nasir and Mahmood [24] concluded that the implementation of flexible schedules allows for the attainment of a healthy balance between work and personal life, decreasing the departure rate of employees, resulting in a greater retention of employees in the organisation.

Hyman and Summers [25] mentioned that work intervenes in workers' personal lives, which can result in emotional exhaustion, giving rise to higher stress rates amongst workers. In this sense, [26] revealed that satisfaction with work–life balance induces a reduction in the occurrence of churning. The following retention strategies are suggested: flexible work schedules, flexible work organisation, provision of adequate resources, adequate workload, and provision of adequate breaks during working hours, while promoting an environment of organisational well-being [27].

### 1.2.5. Salary

Defined as the monetary compensation for the work performed, it is also considered a key factor in maintaining an employee satisfied with the organisation [28]. Ref. [29] also stressed the fact that salary is one of the main stabilising factors of human resources in organisations.

Studies have explored the relationship between compensation and the occurrence of churning. A team of researchers reported that compensation is an important factor in determining whether an employee stays in the organisation [30,31]. In addition, Ref. [32] mentioned that to keep talents in organisations, it is necessary to provide for adequate incentives.

### 1.2.6. Career Progression

Career progression and the opportunity for personal and professional development as a way for workers to achieve their objectives are considered some of the most important factors providing for satisfaction, commitment, and loyalty towards the organisation resulting in their retention [2].

One of the main factors in decreasing the occurrence of churning is the opportunity for personal and professional development. Ref. [15] reported that an important factor in retention is the personal and professional growth of employees; opportunities for promotion increase employee commitment. In turn, Refs. [33,34] emphasised that the opportunity for further learning and development promotes a good work environment and well-being, improving workers' commitment to stay in organisations.

### 1.2.7. Responsibility

The attribution of responsibilities to workers is a motivational factor. It helps develop recognition for one's own value, enhancing trustworthiness for the performance of one's work assigned by the organisation, whether through the attribution of responsibilities in teamwork or through the attribution of responsibilities at an individual level [35].

Johnson [36] in his book "The new rules of engagement" described that one of the biggest organisational battles for the next 10 years will be the ability organisations will have to have in successfully being able to hire workers to stay in business. To this end, it is crucial to fully invest in assigning responsibilities to promote a sense of appreciation and involvement, to provide for equal opportunities, health and safety, and good communication and cooperation, which are some of the factors that promote employee engagement in

organisations, leading to a decrease in the occurrence of churning and promoting retention, high productivity, profits, and returns of investments.

Having defined all main dimensions of the causes of churning of human resources, what follows is the role played by managers of human resources within organisations in minimising the not-so-positive aspects leading to the occurrence of churning, thus helping promote individual and organisational welfare.

*1.3. The Role of Human Resources Management in Mitigating Churning to Promote Individual and Organisational Well-Being*

One of the main challenges organisations are faced with concerns the establishment of organisational policies intending to foster happiness and well-being of their workforce in all aspects of said workers lives. In addition, there is also the need to implement an organisational culture as a system to implement policies, norms, and actions on behalf of their workers to trigger a larger sense of commitment for the achievement of set goals and objectives that will enable the survival and growth of the organisation [37].

Warr [38] referred that work environment influences workers' well-being. It is related to what it can provide for workers in terms of opportunities and positive experiences.

Through several theoretical approaches, the literature focuses on the theme of quality of life in the workplace, which has showed some incongruities and limitations. Taking the above into consideration, the author referred to two perspectives: the organisational perspective, where a set of conditions are adopted to influence workers' well-being, and a second one, emphasising the degree in which workers' individual needs are satisfied [37]. However, it is necessary to differentiate these two perspectives, that is, organisational welfare versus workers' well-being, to distinguish the quality of life experienced by the worker and the conditions of the environment where the worker performs their activities [39].

Walton [40] presented dimensions of the quality of life in the workplace as context variables that in practical terms are the responsibility of the management of human resources, such as the concern with human and environmental values, considering for this purpose eight distinct requirements: a fair and adequate compensation; healthy and safe working conditions; opportunities for skill development; opportunity for continuous growth; social integration in the workplace; constitutionalism; work and living space; social relevance of the life in the workplace.

In view of the above, organisations should adopt strategies that promote well-being that will lead to the decrease in the occurrence of churning of human resources and, thus, reduce the costs associated with the replacements of keyworkers [41].

**2. Materials and Methods**

In accordance with the aforementioned objectives and, considering the theoretical framework, the present empirical research opted for the retraction of quantitative data through an inquiry per questionnaire previously validated, aiming to understand the reality and context experienced by workers of organisations of several sectors of activity in Portugal.

We mention that the questionnaire applied was elaborated and validated by the authors in a previous study; however, due to the fact that it is under evaluation, we cannot cite it.

A sociodemographic questionnaire was used to retrieve data on the sample (age, gender, academic qualifications, function, seniority, contractual relationship, type of work, practiced schedule, salary, type of organisation, sector of activity, and size of organisation). Initially, the questionnaire was composed by 9 dimensions and 35 items, namely, 5 items that make up work environment, 5 items for salary, 4 items for recognition, 3 items for career progression, 5 items for motivation, 4 items for leadership, 3 items for competition, 3 items for schedule, and 3 items for location. Later, through Exploratory Factor Analysis, 2 dimensions (competition and location) were eliminated. In this study, only 7 dimensions and 29 items. Once the study factors were defined, descriptive statistical analysis was used

on the sample using frequency and percentage in the categorical variables (Table 1) and median, mean, and standard deviation in the quantitative variables (Table 2).

**Table 1.** Categorical Variables: Distribution of demographic data of the respondents.

| Variable | Category | Frequency | Percentage |
|---|---|---|---|
| Gender | Male | 371 | 46.3 |
| | Female | 427 | 53.3 |
| | No information | 3 | 0.4 |
| Civil status | Single | 276 | 27 |
| | Married/Non-marital partnership | 491 | 61.3 |
| | Divorced | 90 | 11.2 |
| | Widow(er) | 2 | 0.2 |
| | No information | 2 | 0.2 |
| Academic qualifications | Primary Education | 44 | 5.5 |
| | Secondary Education | 239 | 29.8 |
| | Higher Education | 514 | 64.2 |
| | No information | 4 | 0.5 |
| Function | Direction | 112 | 1 |
| | Management | 118 | 14.7 |
| | Technician | 315 | 39.3 |
| | Administrative Assistant | 93 | 11.6 |
| | Operational | 159 | 19.9 |
| | No information | 4 | 0.5 |
| Antiquity | <2 years | 169 | 21.1 |
| | 2 to 10 years | 289 | 36.1 |
| | 11 to 20 years | 126 | 15.7 |
| | >20 years | 215 | 26.8 |
| | No information | 2 | 0.2 |
| Contractual relationship | Fixed-term contract | 135 | 16.9 |
| | Open-ended contract | 533 | 66.5 |
| | Contract of unspecified duration | 66 | 8.2 |
| | Other | 61 | 7.6 |
| | No information | 6 | 0.7 |
| Type of work | Full time | 784 | 97.9 |
| | Part-time | 10 | 1.2 |
| | No information | 7 | 0.9 |
| Practiced Schedule | Fixed Schedule | 433 | 54.1 |
| | Flexible Schedule | 207 | 25.8 |
| | Shifts | 156 | 19.5 |
| | No information | 5 | 0.6 |
| Salary | EUR 665 | 53 | 6.6 |
| | EUR 700-EUR 1000 | 206 | 25.7 |
| | EUR 1100–EUR 1400 | 216 | 27 |
| | EUR 1500–EUR 1800 | 118 | 14.7 |
| | EUR 1900–EUR 2200 | 50 | 6.2 |
| | EUR 2300–EUR 2600 | 44 | 5.5 |
| | EUR 2700–EUR 3000 | 44 | 5.5 |
| | >EUR 3000 | 63 | 7.9 |
| | No information | 7 | 0.9 |

**Table 1.** *Cont.*

| Variable | Category | Frequency | Percentage |
|---|---|---|---|
| Type of Organisation | National company | 334 | 41.7 |
| | Multinational company | 342 | 42.7 |
| | Public Organisation | 106 | 13.2 |
| | Social Solidarity Institution | 16 | 2 |
| | No information | 3 | 0.4 |
| Sector of activity | Agriculture | 9 | 1.1 |
| | Fishing Industry | 100 | 12.5 |
| | Manufacturing Industry | 217 | 27.1 |
| | Electricity, Gas, and Water Industry | 8 | 1 |
| | Civil Construction and Public Works Industry | 14 | 1.7 |
| | Trade | 38 | 4.7 |
| | Accommodation and Catering Industry | 14 | 1.7 |
| | Transportation and Communications | 25 | 3.1 |
| | Education | 40 | 5 |
| | Business Service | 90 | 11.2 |
| | Healthcare and Social services | 38 | 4.7 |
| | Financial activities | 55 | 6.9 |
| | Other activities | 149 | 18.6 |
| | No information | 4 | 0.5 |
| Dimension of the organisation | <10 workers | 67 | 8.4 |
| | 10 to 50 workers | 105 | 13.1 |
| | 50 to 250 workers | 107 | 13.4 |
| | >250 workers | 518 | 64.7 |
| | No information | 4 | 0.5 |

Source: The authors, based on the data retrieved from the conducted research.

**Table 2.** Quantitative variables to analyse the main causes of churning.

| Factors | Items | N | Median | Mean | Standard Deviation |
|---|---|---|---|---|---|
| Age | Age | 756 | 42 | 41.69 | 10.26 |
| Work Environment | Satisfaction with facilities, equipment, and support services (e.g., parking, occupational health, transportation, sanitary facilities) | 796 | 8 | 7.37 | 2 |
| | Satisfaction with management and colleagues | 800 | 7 | 7 | 2.15 |
| | Satisfaction with socialisation programs (sports, cultural and social activities promoted by the organisation) | 796 | 5 | 4.74 | 2.79 |
| | Comfort and physical well-being in the workplace (e.g., temperature, space, cleanliness) | 799 | 7 | 6.8 | 2.47 |
| | Degree to which one considers that there is a conflict-free work environment in the organisation | 795 | 7 | 6.32 | 2.41 |

**Table 2.** *Cont.*

| Factors | Items | N | Median | Mean | Standard Deviation |
|---------|-------|---|--------|------|--------------------|
| Salary | Feeling that one is being well-paid | 800 | 6 | 5.34 | 2.66 |
| | Feeling that one's remuneration is fair and equitable regarding one's peers in a similar situation within the organisation (e.g., performed functions, seniority, qualifications, and performance) | 797 | 4 | 4.76 | 2.66 |
| | Salary as a decisive factor for permanence in the organisation | 800 | 7 | 6.25 | 2.49 |
| | Satisfaction with perks and benefits granted by the organisation (e.g., protocols, insurance, vacations, sickness leaves, etc.) | 796 | 7 | 6.43 | 2.42 |
| | Desire to leave the organisation due to remuneration aspects | 799 | 4 | 4.68 | 2.83 |
| Recognition | How the organisation recognizes all the work, dedication and effort showed by the employee | 798 | 5 | 5.29 | 2.74 |
| | Frequency in which management praises a job well done | 797 | 6 | 5.41 | 2.89 |
| | Regularly obtaining information about your performance in the organisation | 795 | 5 | 5.42 | 2.74 |
| | Degree of satisfaction with performance evaluation | 792 | 6 | 5.79 | 2.8 |
| Career Progression | Opportunities for professional progression in the organisation | 796 | 5 | 4.98 | 2.86 |
| | Satisfaction with vocational training received | 798 | 5 | 5.31 | 2.8 |
| | Feeling that employees who are promoted are those who effectively demonstrate the best performance and potential to take on that role | 791 | 5 | 4.89 | 2.78 |
| Motivation | Overall satisfaction with the organisation, considering all the experience of working there | 796 | 7 | 6.22 | 2.33 |
| | Feeling that one works in a solid organisation and with prospects for the future | 797 | 7 | 6.59 | 2.52 |
| | Prestige for being a constituent of the organisation | 796 | 7 | 6.42 | 2.61 |
| | Personal fulfilment for the role assigned in the organisation | 797 | 7 | 6.38 | 2.58 |
| | Feeling that assigned objectives are challenging | 793 | 7 | 6.27 | 2.61 |
| Leadership | Autonomy that is given to plan, execute and evaluate one's own work | 793 | 7 | 6.76 | 2.61 |
| | Efficiency of working planning methods in your Direction/Department | 798 | 6 | 5.91 | 2.55 |
| | Access to and receiving of information which one considers useful for the performance of one's function | 798 | 7 | 6.32 | 2.43 |
| | Support one can count on the part of superiors | 798 | 7 | 6.64 | 2.79 |

**Table 2.** *Cont.*

| Factors | Items | N | Median | Mean | Standard Deviation |
|---------|-------|---|--------|------|--------------------|
| Schedule | Degree of satisfaction with the practiced schedule | 798 | 8 | 6.81 | 2.68 |
| | Degree of satisfaction with personal and professional life conciliation | 793 | 7 | 6.41 | 2.68 |
| | Possibility to adjust working hours according to workers needs | 797 | 7 | 6.32 | 2.99 |

Source: The authors, based on the research data.

In order to quantify the measurement variable related to the causes of churning, participants had to respond to each of the statements according to their professional reality on a 10-point Likert scale (Very low = 1; Very high = 10). According to [42], a Likert scale of 10 points ensures that the distribution of results around the average is more diffuse, with a broader distribution, providing greater discriminating power and, consequently, a more reliable ability to isolate good and bad performances.

### 2.1. Procedures

#### 2.1.1. Pre-Test

Before proceeding to data collection, a pre-test stage was applied, testing the data collection and the application of the questionnaire through the LinkedIn-Corporation social network, in order to assess the feasibility and relevance of the data, as well as the way the questionnaire was constructed and structured.

For the pre-test, a convenience sampling of 10 respondents was used.

Based on the application of the pre-test, it was possible to identify problems or difficulties in the interpretation of the questionnaire, whether these were related to the formulation of the text, the format, or the response time in order to identify the need to modify some points.

The pre-test stage revealed that no changes were needed to the initial version of the questionnaire, so it was fully used in the subsequent phase of data collection.

#### 2.1.2. Data Collection

The questionnaire was elaborated through the platform Google Forms and applied on the social network LinkedIn-Corporation given that it is a professional network which makes it possible to reach a more diverse population in terms of sectors of activity and of professional backgrounds. The questionnaire was applied between June and September of 2021.

Initially, as a method of selecting respondents, we looked for workers from various professional areas and sectors of activity and sent a message calling for their participation through a link generated by the Google Forms platform.

When acceding the questionnaire, the participants had access to the objectives of the study. In it, anonymity and confidentiality were assured and instructions for its completion were also provided. It was also requested the indication of the degree in accordance with each affirmation with resource to a Likert scale of 10 points, with 1 corresponding to the lowest degree possible and 10 to the highest. An average time of approximately 20 min was estimated for its completion.

### 2.2. Methodological Options in the Treatment and Analysis of Data

We can define the methodology as the totality of systematic and rational activities leading to the achievement of the proposed objectives, with the maximum safety and economy of effort and time, as well as other resources, in order to allow the delineation of the path to be followed, enabling the detection of errors and simultaneously assisting the researcher throughout their decision-making process [43].

To analyse the variable under discussion, a multivariate statistical analysis was adopted by initially resorting to an Exploratory Factor Analysis (EFA), given that, through this analysis, it is possible to explore data, allowing information to be provided to the researcher on how many factors are necessary to better represent the data, that is, the EFA can be used to supply a preliminary verification on the number of factors and the standard loads. Items with lower loads are candidates for elimination [44].

Finally, we resorted to a TwoStep Cluster analysis, aimed at disclosing natural groupings (Clusters) based on a set of previously selected variables to later proceed to the analysis of the intrinsic characteristics from each of these groups that would not be apparent otherwise, which is also an applicable method to the samples of any dimension [45].

### 2.3. Sample

The sociographic variables used and identified within the questionnaire were: age, gender, civil status, and academic qualifications. Regarding occupational characterisation, the following were identified: function, antiquity, contractual relationship, type of work, practiced schedule, salary, type of organisation, sector of activity, and the dimension of the organisation.

Tables 1 and 2 present the totality of variables used in the questionnaire allowing the necessary overall information on the theme of the dimensionality of churning to be obtained. It should be noted that of the 42 variables designed for this study, 12 are categorical (Table 1) and 30 are quantitative (Table 2). The totality of these variables will make it possible to meet the specific objectives mentioned above.

The sample was selected by convenience, where the participants were workers of organisations from several sectors of activity. Table 1 shows the distribution of the demographic data of the respondents.

Through the results gathered, it was possible to make a global analysis of the studied sample (Tables 1 and 2). The sample was composed of 801 participants, of which: 46.3% were male and 53.3% were female, aged between 20 and 74 years old, mostly married or in a non-marital partnership (61.3%), and had higher educational academic qualifications (64.2%), from which 39.3% were technicians, 19.9% had operational positions, 14.7% had managerial positions, 14% were directors, and 11.6% were administrative assistants. As for seniority in the organisation, the highest percentage of workers remained within the organisation for 2 to 10 years (36.1%), enjoying an open-ended contract (66.5%); 16.9% had a fixed-term contract, 8.2% had an unspecified-duration contract, and 7.6% were self-employed. As for working time, 97.9% worked full-time and 1.2% worked part-time, 54.1% practiced a fixed schedule, followed by a flexible schedule (25.8%), and 19.5% worked in shifts, with a large percentage of workers receiving a salary between EUR 1100 and EUR 1400 (27%).

Regarding the type of organisation, 42.7% worked in multinational companies, 41.7% worked in national companies, 13.2% worked in public administration organisations, and 2% in social solidarity institutions. Regarding the activity sector, it was ascertained that most workers worked in the manufacturing industry sector (27.1%) in organisations employing more than 250 workers (64.7%).

Through the statistical analysis with recourse to the Exploratory Factor Analysis (EFA), the factor competition presented one alpha of Cronbach = 0.159. In the matrix of correlations between items, all had presented inferior values to 0.5, hence opting for the exclusion of said factor. As for the locality factor, it presented negative values in Cronbach's alpha (−0.499) due to a negative mean covariance between items, which invalidates the assumptions of the reliability model. In the matrix of correlations between items, it presented negative values; hence, we resorted to its reverse; however, the matrix of correlations continued to present negative values between the items; thus, it was also excluded. From the 9 initial factors, only 7 were considered (Table 2).

After performing the analysis regarding the satisfaction of workers with labour conditions, and taking into consideration the selected factors, namely: Work Environment; Salary;

Recognition; Career Progression; Motivation; Leadership; Schedule, it was ascertained that in terms of the factor Work Environment, workers valued satisfaction with facilities, equipment, and support services (on average, 7.37); with regard to the factor Salary, workers reported satisfaction with benefits and benefits granted by the organisation (on average, 6.43); in terms of the factor Recognition, workers valued the degree of satisfaction with performance assessments (on average, 5.79); regarding the Career Progression factor, workers valued satisfaction with vocational training received (on average, 5.31); regarding the Motivation factor, workers valued the feeling that they work in a solid organisation and with prospects for the future (on average, 6.59); in terms of the Leadership factor, workers valued autonomy given to plan, execute, and evaluate their own work (on average, 6.76); in terms of the Schedule factor, workers valued satisfaction with working hours practiced (on average, 6.81).

## 3. Results

*Analysis and Discussion of Results*

From the 42 designed variables for the inquiry per questionnaire, in the TwoStep Cluster analysis, 5 quantitative variables were considered, 2 in reference to recognition, 2 referring to leadership, and 1 to motivation (N = 801). The option for these variables rested upon the highest values in reference to the level of importance.

The TwoStep Cluster analysis organised individuals in two groupings, where the median evaluations of the variables considered in the research model are observed in Table 3 and Figure 1. In addition to the global perceptions of the medians for each variable, it was possible to verify the difference obtained by the TwoStep Cluster analysis regarding the median evaluations attributed by the respondents belonging to each of the groups in their responses to the survey (Figure 1).

**Table 3.** Clusters to assess the vulnerability of churning.

| | Frequency with Which Management Praises a Job Well Done | How the Organisation Recognizes All the Work, Dedication and Effort Made by the Employee | Efficacy of Working Planning Methods in One's Direction/Department | Overall Satisfaction with the Organisation, Considering all the Experience of Working in It | Access to and Receiving of Information Which One Considers Useful for the Performance of One's Function |
|---|---|---|---|---|---|
| Global (median) | 6 | 5 | 6.01 | 6.99 | 6.99 |
| Cluster 1- Workers in churning | 2.99 | 2.99 | 4.01 | 4.99 | 4.99 |
| Cluster 2- Enthusiastic workers | 8 | 8 | 8.01 | 8.01 | 8.01 |

Source: The authors.

Based on these findings, Table 3 proposes what can be expected in terms of inequalities in satisfaction for remaining in the organisation by workers belonging to the two Clusters identified in the analysis, with particular emphasis on the possibility to establish an association between workers in churning (Cluster 1) and enthusiastic workers (Cluster 2).

Of all variables mentioned in Table 3, enthusiastic workers (Cluster 2) are more prone to remain in the organisation given that a superior median in all variables was verified when comparing it to its global median. This was ascertained in all variables (median 8.01) except for the following: frequency with which management praises a job well done; the way an organisation recognises all work performed, dedication, and efforts made by workers (median 8). Cluster 1 presents a median inferior when in comparison with the global median. The Cluster related to workers in churning was evidenced in the following variables: frequency with which management praises a job well done; how an organisation recognises all work performed, dedication, and efforts made by workers (median 2.99); efficacy of the working planning methods in one's direction/department (median 4.01); global satisfaction with the organisation considering all working experience within the

latter; access to and receiving of information one considers useful for performing one's functions (median 4.99).

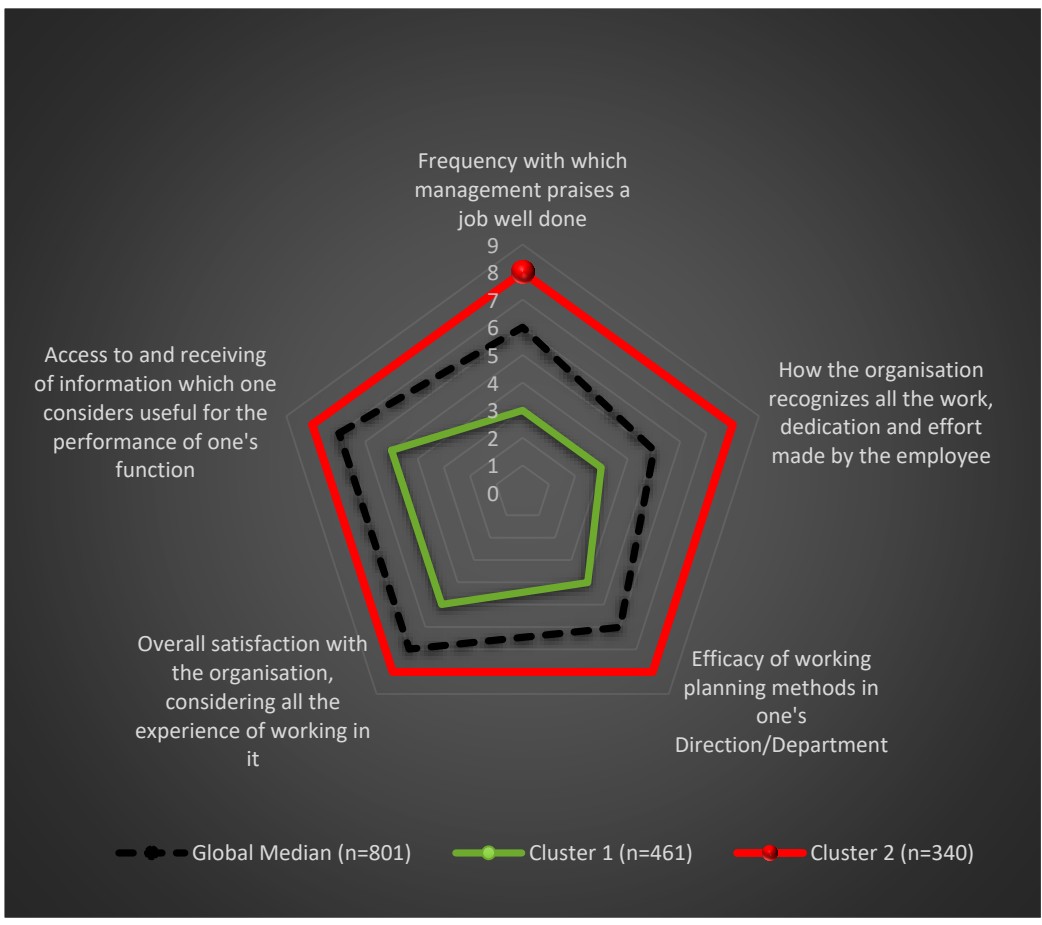

**Figure 1.** Comparative graphic of the medians of the variables for the various Clusters and for the globality of respondents. Source: The authors.

Cluster 1 (composed by 461 observations, which correspond to 57.5% of the total) is constituted by workers in churning. The latter is a group of workers mainly constituted by women, with a median age of 47 years old, married, possessing a higher education, working in multinational companies belonging to the manufacturing industry with a dimension superior to 250 workers. They perform technical functions, have been working in said organisations for 2 to 10 years, have a permanent employment contract, work full-time, and have a fixed schedule. Salary-wise, the largest percentage of respondents earned around EUR 700 and EUR 1000.

Cluster 2 (composed of 340 observations, which correspond to 42.4% of the total) is constituted by enthusiastic workers. It is a group of workers essentially constituted by women, with median age of 40 years old, married, possessing a higher education, working for national companies in non-specified sectors of activity (in the questionnaire, this was classified as "other activities"), in companies whose dimensions are superior to 250 workers. They perform technical functions, have a seniority in the organisation between 2 and 10 years, have a permanent employment contract, work full-time, and have a fixed schedule. Salary-wise, there is a higher percentage of respondents earning between EUR 1100 and EUR 1400.

### 4. Final Conclusions, Limitations of the Study, and Future Studies

The present study aimed to identify the dimensions related to the causes of churning and to analyse its applicability in the management of human resources to provide for individual and organisational welfare.

It intended to make a sociographic characterisation of workers, as well as their professional characterisation and, finally, to analyse the perception workers have regarding all dimensions addressed in the present research.

Resorting to the literature review, it was possible to contextualise the problem under study about churning of human resources, as well as to ascertain the role the management of human resources has in promoting individual and organisational welfare.

On the basis of the theoretical framework and in accordance with the targets set, quantitative data were retrieved using a questionnaire survey that, through the analysis of its results, allowed a global analysis of the categorical variables (Table 1) and of the quantitative variables (Table 2), allowing for the workers of the organisations under study to be globally categorised, which enabled the defined objectives to be answered.

Through the Exploratory Factor Analysis (EFA), it was possible to select the most relevant dimensions to be considered; afterwards, a characterisation of workers and of the items under study was performed. Finally, through the TwoStep Cluster Analysis, carried out through the sample of workers regarding recognition, leadership, and motivation (having this selection been made given the highest levels of significance), two Clusters could be identified (Table 3 and Figure 1): workers in churning (Cluster 1) and enthusiastic workers (Cluster 2).

Resulting from the TwoStep Cluster analysis, it was possible to distinguish which were the workers pertaining to each of the Clusters. It was evidenced that both Clusters were composed by women as, in Cluster 2, the level of permanence in the organisation was more expectable given that it presented a greater satisfaction with labour conditions in comparison to Cluster 1.

However, there was a significant difference. It was assumed that the main reason for such a difference regarding working conditions is related to salary, given that workers in Cluster 1 are workers who had lower wages when compared to workers from Cluster 2.

It is suggested that for future studies, one should replicate the previously described approach in order to strengthen the aforementioned conclusions. As for the method of analysis, we propose confirmatory factor analysis (CFA) to reach the causal model that allows the application of structural equations. These new results would be extremely relevant if considered in other countries given the chance that they would allow the undertaking of a crossing and comparison of the results retrieved in this study regarding the issue of churning of human resources.

#### 4.1. Study Contributions and Limitations

This study was conducted given the scarcity of literature and empirical studies presently available, as well as the relevancy in operationalising a concept still little disseminated in Portugal, that is, churning of human resources.

Through the present research, it was possible to gain further knowledge on a greatly complex and relevant subject in the area of human resources, to ascertain the perception of human resources (belonging to Portuguese organisations from several sectors of activity), allowing for a true understanding in regard to the reality of the organisational context of the subject at hand.

This study had some limitations, namely the scarcity of existing literature on the topic; the fact that the questionnaire applied was developed and validated by the authors, meaning that it cannot be cited, due to being in the process of evaluation in another journal; another limitation concerned the fact that only one method of data collection was applied, which could enrich the study through the application of interviews and confront the results obtained.

Theoretical and Practical Contributions

This study allowed for the expansion of the theoretical framework through resorting to the approach of the theme of churning, the selection of the dimensions of the causes of churning, and the role played by the management of human resources in the attenuation of churning of human resources for the promotion of individual and organisational welfare.

In terms of its applicability for the management of human resources, through the retrieved results, it was possible to elaborate and implement more sustainable and effective practices and policies of human resources to promote organisational welfare to provide for workers' satisfaction, contributing to their permanence in organisations and, thus, contributing to the decrease in the occurrence of churning.

In terms of contributions for organisational practices, this study entails the ability to undertake a survey of needs or the improvement of existing working conditions.

Finally, it is expectable that this study can contribute to a shift in organisational paradigms, through a preventive model of the causes of churning to promote better working conditions, and, in such a way, to contribute for the reduction in the percentage of churning and to reduce costs with the unexpected substitutions of workers. Taking the above into consideration, through the implementation of predictive models of churning of human resources, it is expected that organisations will have the opportunity to deepen their understandings regarding the causes of churning, enabling the implementation of strategies and retention plans through the improvement of team management and organisational policies [46].

**Author Contributions:** Conceptualization, O.A.C.P. and P.M.A.R.C.; methodology, O.A.C.P. and P.M.A.R.C.; validation, O.A.C.P. and P.M.A.R.C.; formal analysis, O.A.C.P. and P.M.A.R.C.; investigation, O.A.C.P. and P.M.A.R.C.; resources, O.A.C.P. and P.M.A.R.C.; writing—original draft preparation, O.A.C.P.; writing—review and editing, O.A.C.P. and P.M.A.R.C.; visualization O.A.C.P. and P.M.A.R.C.; supervision, P.M.A.R.C.; project administration, P.M.A.R.C. All authors have read and agreed to the published version of the manuscript.

**Funding:** This work is supported by Portuguese national funds through FCT—Fundação para a Ciência e a Tecnologia, under project UIDB/04643/2020.

**Conflicts of Interest:** The authors declare no conflict of interest.

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
