# Peer review of "Dimensionality of the Causes of Churning: A Multivariate Statistical Analysis"

_merits, doi:10.3390/merits3010002_

Round 1
Reviewer 1 Report (Previous Reviewer 1)
The paper now received is almost a completely new paper with empirical data. This fact, in my view, makes the paper more interesting to readers.
My comments will address the differences found regarding the original paper.
Comments on the major sections of the paper:
Introduction: This section was improved with a clearer statement on the purpose of the paper and method used in the empirical study.
Materials and methods: The method and data gathering are suitable for this type of research. However, it would be useful to know the origin of scales/items, based on existing literature or self-developed by authors?
Analysis and presentation of results: The statistical procedures used for analysis are acceptable in view of the objectives; findings are presented in a clear and coherent manner.
Final conclusions, study limitations and future studies: Conclusions are in line with the findings and future steps in empirical research are mentioned, but no acknowledgment of limitations is made, and we believe that it should be included.
Author Response
Reviewer 1
Dear reviewer, first of all we thank you for your valuable suggestions, which required our full consideration as well as the opportunity to improve our article, allowing us to enrich our scientific knowledge.
Comments, Suggestions for Authors and reply/justification to reviewers:
The paper now received is almost a completely new paper with empirical data. This fact, in my view, makes the paper more interesting to readers.
My comments will address the differences found regarding the original paper.
Comments on the major sections of the paper:
Introduction: This section was improved with a clearer statement on the purpose of the paper and method used in the empirical study.
Materials and methods: The method and data gathering are suitable for this type of research. However, it would be useful to know the origin of scales/items, based on existing literature or self-developed by authors?
(The scale applied was developed and validated by the authors in a previous study, however, as it is under evaluation process, we could not place the citation).
Analysis and presentation of results: The statistical procedures used for analysis are acceptable in view of the objectives; findings are presented in a clear and coherent manner.
Final conclusions, study limitations and future studies: Conclusions are in line with the findings and future steps in empirical research are mentioned, but no acknowledgment of limitations is made, and we believe that it should be included.
(limitations of the study were added as suggested).
We thank you for your valuable appreciation and availability to our article.
Best regards

Reviewer 2 Report (New Reviewer)
While this study construct is quite good. The authors are encouraged to rewrite this paper and resubmit it later. Here are some critical comments:
- Line 125 and many similar problems: Please read other published articles to know how to cite correctly.
This paper need to be significantly check the format of the words (equal size,...).
- Please clearify the methods that you use in this paper. Why you choose such methods but not the others?
How EFA and many other methods works?
How can you validate the results?
- Please translate the meaning of the findings into the real word (specify research context).
Author Response
Reviewer 2
Dear reviewer, first of all we thank you for your valuable suggestions, which required our full consideration as well as the opportunity to improve our article, allowing us to enrich our scientific knowledge.
Comments, Suggestions for Authors and reply/justification to reviewers:
While this study construct is quite good. The authors are encouraged to rewrite this paper and resubmit it later. Here are some critical comments:
- Line 125 and many similar problems: Please read other published articles to know how to cite correctly.
(The citations were reviewed taking into account the standards of the journal as well as the use of other articles already published)
This paper need to be significantly check the format of the words (equal size,...).
(as suggested, the format, font size, and table formatting were revised according to the template made available by the journal)
- Please clearify the methods that you use in this paper. Why you choose such methods but not the others?
(The answer to this question is explained in point 2.3 Methodological Options in the Treatment and Analysis of Data)
How EFA and many other methods works?
(In this specific study, we chose these methods, as they were considered to be the most appropriate to achieve the study objectives. In future studies, we expect to use the exploratory factor analysis and the confirmatory factor analysis to reach the causal model that allows for the application of structural equations. This is not the intention of this study, since we did not apply any model to analyse the relationship between the variables).
How can you validate the results?
- Please translate the meaning of the findings into the real word (specify research context).
(Perhaps you mean turnover, however, although turnover is related to churning, it is not our intention to work on the concept of turnover, but to operationalize, disseminate and expand the literature on the concept (theme) of human resources churning, through this empirical study conducted in Portugal).
We thank you for your valuable appreciation and availability to our article.
Best regards

Reviewer 3 Report (New Reviewer)
The article presents the results of research with the aim identification of the dimensions related to the causes of churning and to analyze its applicability in the management of human resources to promote individual and corporate welfare.The authors rely on a wide range of literature,used appropriate methods and a sufficient sample of respondents in the research. Research has brought new interesting knowledge in this area.
Author Response
Reviewer 3
Dear reviewer, first of all we thank you for your valuable suggestions, which required our full consideration as well as the opportunity to improve our article, allowing us to enrich our scientific knowledge.
Comments, Suggestions for Authors and reply/justification to reviewers
The article presents the results of research with the aim identification of the dimensions related to the causes of churning and to analyze its applicability in the management of human resources to promote individual and corporate welfare.The authors rely on a wide range of literature,used appropriate methods and a sufficient sample of respondents in the research. Research has brought new interesting knowledge in this area.
We thank you for your valuable appreciation and availability to our article.
Best regards

Round 2
Reviewer 2 Report (New Reviewer)
Merit is not listed in high ranking list of journals.
However, I hope that the authors can improve the quality of their paper.
Please be in mind that the quality of this paper is not good enough. There are many weakness.
I hope the authors can improve their ability and publish more quality papers soon.
This manuscript is a resubmission of an earlier submission. The following is a list of the peer review reports and author responses from that submission.
Round 1
Reviewer 1 Report
General considerations:
We believe the paper is within the scope of the journal. The topic is interesting, the writing is adequate and concise. The title, abstract, and main sections of the paper are consistent with the literature review presented. The global aims are clearly stated, the analysis conducted is descriptive in line with the objective and methodology. The conclusions seem supported by data presented. The references also seem adequate and up to date.
Comments on the major sections of the paper:
Introduction: The authors introduce the concept of churning, acknowledge its relationship with turnover and why it’s distinct, and explain the importance research on this topic for HRM field.
Materials and methods: This paper is a literature review, so the search for academic papers on the specific topics of interest on major databases seems appropriate.
Analysis and presentation of results: In this section the findings of the literature review are presented in a coherent manner with the respective references. I would add that these findings on churning are also in line with the literature on turnover and retention.
Conclusions: The conclusions are consistent with the results previously presented.
Final conclusions, study limitations and future studies: The authors recognize the limitations such as the lack of empirical studies on the subject, which limits the literature review and inform the reader that this is the first step for an empirical study.
Reviewer 2 Report
The biggest issue with this paper is that the subject and scope of the study is limited. This study uses the term 'churning' but in fact, many researchers have been conducted a vast of research under the name of 'employee voluntary turnover'. There have been several review papers and meta-research on this in the past 40 years (see Cotton & Tuttle, 1986, Academy of Management Review; Griffeth et al., 2000, Journal of Management; Heavey et al., 2013, Journal of Applied Psychology), and hundreds of empirical papers have been conducted on how to predict and prevent turnovers. Even if the same phenomenon is expressed in different terms (i.e., turning), you must refer to the ‘voluntary turnover’ papers to write a paper on the same phenomenon. This study was written by reviewing 20 papers, which is far less than the total amount of published academic papers. Therefore, it is difficult to say that it is a scientifically valid approach.
Reviewer 3 Report
Churning in Human Resources is an interesting topic. The innovation of the topic is relatively good. But, unfortunately, the attitude of writing is sloppy. For example, 4.2.1. Theoretical and Practical Contributions and 5.1.Study Contributions are almost duplicates. Although the paper claims to have retrieved publications between 2011 and 2020, these references are old. I did not see the latest literature in the last three years. In addition, the format of the references is not standard, and the references are not listed according to the requirements of our journal. This reduces the originality of the paper.